# Position-Adaptive Optimization for Agent Confrontation Using the Game Triangle Plane Model

Xiangri Lu[a,b,*]

[a]Beijing Institute of Technology, Haidian District, Beijing, PRC

[b]National Key Lab of Autonomous Intelligent Unmanned Systems, Haidian District, Beijing, PRC

**Abstract**

Addressing the challenge of requiring extensive data and computational resources in intelligent control game systems, this paper proposes a triangular game relationship model in a plane that includes dynamic player A, dynamic player B, and the fixed-point base camp b of player B. This model avoids the use of information shielding technology in real game scenarios, where only small-scale effective data can be obtained to support the operation of intelligent game control systems. In this triangular game relationship model, the coordinate position relationships between player A, player B, and the base camp b of player B provide the constraints conditions for optimizing confrontation positions. Subsequently, the objective function for player A's optimal game position is constructed, and the local optimal confrontation position is obtained and verified by combining the optimization constraints.

Keywords:Intelligent control game;Game Triangle;Adversarial Positioning Optimization;constraint condition;constraint conditions;Objective function

## 1.Introduction

In wargaming simulations, if both opposing parties want to analyze decision-making processes using intelligent control game methods, substantial data support or external model intervention is required. In recent years, large-scale data wargaming platforms such as CMANO Commander, Mencius Joint Operations Simulation Platform, and Combat Brain System have emerged[1-3]. Wargaming simulations, based on incomplete information, are adversarial games where data acquisition is difficult, and different types of models have varying requirements for data volume[4]. To address the issue of insufficient data preventing wargaming

systems from meeting simulation requirements in real-time, using externally established models to support the system will be a beneficial supplement to large-scale data-supported models.

In the 1960s and 1970s, adaptive network models were applied to the decision-making process in war games. The adaptive networks in war game simulations adjusted their structure and parameters according to changes in the environment and their own state to adapt to different tasks and requirements. However, these models could only obtain a limited number of data samples and required a large amount of computational resources[5].Since the 1980s, model predictive control for war games has been able to handle multivariable nonlinear systems. Nevertheless, the accuracy and precision of these models can be affected by various factors [6,7].Since the 2020s, war game simulations have been combined with deep learning models for application in adversarial games, where both sides are in a state of information blackout, making it increasingly difficult to obtain complete information about the opponent. The lack of data input during model training has resulted in a significant waste of computational resources[8,9].

When the small-scale data is ready, the built model will adaptively substitute a small amount of known data (such as player A coordinates, obstacle coordinates of player B arrangement, and coordinates of the base camp b of player B coordinates) into the build constraint target function, thus automatically adjusting the position of player A attack.

2 Optimization of agent counter position coordinates

2.1 Constructing an Confrontation Two-Player Game Environment

Assuming that the two units are playing games on the two-dimensional plane, let's suppose that there is the player A, and the player B sets up obstacles at two points (which can be ignored in terms of size) to construct each other. In a $x_{unit} \times y_{unit}$ area( $x_{unit}$ and $y_{unit}$ are each a unit length arbitrarily.)Construct a two-dimensional game environment in the plane. This article sets the starting point of the player A as the red mark point, and the player A looks for the base camp b of player B, and the player B that sets up obstacles to prevent Player A from attacking

the base camp b of player B is marked in green, and the base camp b of player B which is a fixed point $(x_{goal}, y_{goal})$ in the two-dimensional game plane is marked in blue. At this point, This article construct a model that a game triangle in the two-dimensional game plane with the player A and the obstacles arranged by the player B and the base camp b of player B , as shown in Figure 1.

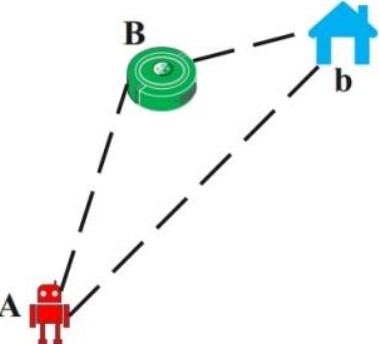

Figure 1: The Composition of Game Participants in a Two-dimensional Game Plane Forms a Game Triangle.

2.2 Describe the Position of the Game Triangle

In a confrontation game environment, the player A approaches the base camp b of player B, seeking to find the optimal attacking position. According to the description of the confrontation game environment, the process of confrontation game is represented by the player A,the base camp b of player B and the player B forming a game triangle. Within the two-dimensional confrontation game plane, the player A finds the attacking position closest to the base camp b of player B, while the player B tries to get as close as possible to its base camp b to protect it. At this point, the player A needs to approaching the base camp b of player B while maintaining a certain safe distance from the player B. Assuming an extreme scenario where the player B and the base camp b of player B are infinitely close, the player A in the two-dimensional confrontation game plane would be the optimizing attack coordinates in relation to the base camp b of player B.

In the process of confrontational game with incomplete information, it is assumed that the player A and the coordinates of the player B are respectively $A(x_1, y_1)$ , $B(x_2, y_2)$ , and $b(x_{goal}, y_{goal})$ simultaneously form a game triangle. The three points A, B, and b constitute a schematic diagram of the position

relationship of the game triangle, as shown in Figure 1. Then this game triangle has the following relationship in the Offensive and Defensive confrontation Game.

I.Positional relationship

According to the schematic diagram of the position relationship of the game triangle shown in Figure 1, when the player A obtains the best weaken position, the strategy during the game is that neither the player A nor the player B has too much contact with the other party to consume each other, and both parties must be closest to the base camp b of player B (player B can enhance protection, and the player A can increase the ability to weaken the base camp b of player B). That is, the positional relationship between the player A,the player B and the base camp b of player B is as shown in Equations (1) and (2).

$$\vec{BA} + \vec{Ab} = \vec{Bb} \tag{1}$$

$$\vec{Bb} \perp \vec{Ab} \tag{2}$$

II. The relative positional trajectory relationship between player A and the fixed base point b.

If the player A's position conforms to Gaussian distribution in the two-dimensional game environment plane (also known as a normal distribution, a common continuous probability distribution,Gaussian distribution is often used to represent an unknown random variable in statistics)[10,11].At this point, the position of player A randomly moves in a two-dimensional game environment plane. Our objective is to identify the optimal attack distance between player A and the base camp b of player B within the plane. This involves determining the quadratic form of the Euclidean distance between player A and the base camp b of player B in a Gaussian distribution. In this paper, the optimal attack distance of player A is inferred by setting the probability of player A near the fixed point of the base camp b of player B. For interpretation, we decompose the trajectory vector of player A in the adversarial game into two directions, x and y. Meanwhile, the motions in both directions of x and y are independent of each other. Here, this paper first performs a general derivation, assuming that the motion vector of player A is A multivariate variable.

$$A \sim N(\mu,\Sigma) = \frac{1}{(2\pi)^{\frac{P}{2}}|\Sigma|^{\frac{1}{2}}}\exp(-\frac{1}{2}(A-\mu)^T\Sigma^{-1}(A-\mu)) \tag{3}$$

$$A \in \Re^P, r.v., A = (A_1, A_2, ..., A_p)^T, \mu = (\mu_1, \mu_2, ..., \mu_p)$$

$$\Sigma = \begin{pmatrix} \sigma_{11} & \sigma_{12} & ... & \sigma_{1p} \\ \sigma_{21} & \sigma_{22} & ... & \sigma_{2p} \\ ... & ... & ... & ... \\ \sigma_{p1} & \sigma_{p2} & ... & \sigma_{pp} \end{pmatrix}_{p \times p}$$

and satisfied $\Sigma = U\Lambda U^T, \Lambda = diag(\lambda_i), i = 1,2,...,p$ , $UU^T = U^TU = I$ ,so

$$\Sigma = U\Lambda U^T = (U_1, U_2, ..., U_p)\begin{pmatrix} \lambda_1 & 0 & ... & 0 \\ 0 & \lambda_2 & ... & 0 \\ ... & ... & ... & ... \\ 0 & 0 & ... & \lambda_p \end{pmatrix}\begin{pmatrix} U_1^T \\ U_2^T \\ ... \\ U_p^T \end{pmatrix} \tag{4}$$

$$\Sigma = (U_1\lambda_1, U_2\lambda_2, ..., U_p\lambda_p)\begin{pmatrix} U_1^T \\ U_2^T \\ ... \\ U_p^T \end{pmatrix} = \sum_{i=1}^{p} U_i\lambda_i U_i^T \tag{5}$$

$$\therefore \Sigma^{-1} = (U\Lambda U^T)^{-1} = (U^T)^{-1}\Lambda^{-1}U^{-1} = U\Lambda^{-1}U^T = \sum_{i=1}^{p} U_i\frac{1}{\lambda_i}U_i^T \tag{6}$$

$$\therefore \exists\forall\Delta = (A-\mu)^T\Sigma^{-1}(A-\mu) = (A-\mu)^T\sum_{i=1}^{p} U_i\frac{1}{\lambda_i}U_i^T(A-\mu) \tag{7}$$

$$\because \forall m = \begin{pmatrix} m_1 \\ m_2 \\ ... \\ m_p \end{pmatrix}, m_i = (A-\mu)^T U_i$$

$$\Delta = \sum_{i=1}^{p}(A-\mu)^T U_i\frac{1}{\lambda_i}U_i^T(A-\mu) \tag{8}$$

$$\Delta = \sum_{i=1}^{p} m_i\frac{1}{\lambda_i}m_i^T = \sum_{i=1}^{p}\frac{1}{\lambda_i}m_i m_i^T \tag{9}$$

Moreover, since the player A, the player B, and the base camp b of player B are all within the confrontational game plane, so $p = 2$ . The coordinates of the anti-interference game strategy at the player A satisfy equation (10).

$$\begin{cases} p(x) = \dfrac{1}{(2\pi)|\Sigma|^{\frac{1}{2}}} \exp(-\dfrac{1}{2}\Delta) \\ \dfrac{m_1^2}{\lambda_1} + \dfrac{m_2^2}{\lambda_2} = \Delta \end{cases} \tag{10}$$

Here we assume that the probability of player A appearing near the fixed point b is 0.9. To simplify the calculation, the random motion variables of the player A in the x and y directions are set as standard normal distributions.So $\Delta = -3.47$.

Ⅲ. The Relative Position Relationship between the player A and the player B

When the player A and the player B's game tends towards stability, the relative distance between the two should remain unchanged. So $\lim\limits_{t\to\infty} \left\| \overrightarrow{AB} \right\|_2 = r$ .As shown in Equation (11).

$$(x_1 - x_2)^2 + (y_1 - y_2)^2 = r^2 \tag{11}$$

In summary, in the incomplete information confrontation game process, the player A, the player B coordinates are respectively, $A(x_1, y_1)$ , $B(x_2, y_2)$ and $b(x_{goal}, y_{goal})$ form a triangular game. The relationship between the player A, the player B, and the base camp b of player B is as shown in Equation (12).

$$\begin{cases} \overrightarrow{BA} + \overrightarrow{Ab} = \overrightarrow{Bb} \\ \overrightarrow{Bb} \perp \overrightarrow{Ab} \\ \dfrac{m_1^2}{\lambda_1} + \dfrac{m_2^2}{\lambda_2} = \Delta \\ (x_1 - x_2)^2 + (y_1 - y_2)^2 = r^2 \end{cases} \tag{12}$$

$$\Rightarrow \begin{cases} x_1 x_2 + y_1 y_2 - x_{goal}(x_1 + x_2) - y_{goal}(y_1 + y_2) + x_{goal}^2 + y_{goal}^2 = 0 \\ \dfrac{m_1^2}{\lambda_1} + \dfrac{m_2^2}{\lambda_2} = \Delta \\ x_1^2 + x_2^2 + y_1^2 + y_2^2 - 2(x_1 x_2 + y_1 y_2) = r^2 \end{cases}$$

3. The Optimal Weaken Position of the player A

3.1 The local optimal solution of the agent

Since it is an incomplete information game of confrontation process, where the player A and the player B are not know each other's specific location information. The strategy of the player A is to stay away from the player B and get as close to the base camp b of player B as possible. Similarly, the player B also needs to stay away from

the player A while getting as close to the base camp b of player B as possible. According to this description, the positional relationship between the the player A and the player B in the confrontation game can be expressed as shown in formula (2),so $x_1 x_2 + y_1 y_2 = 0$.

So, under this war game, the two constraint conditions are as shown in formulas (14), (15), (16), and (17).

$$x_1^2 + x_2^2 + y_1^2 + y_2^2 = r^2 \qquad (14)$$

$$\frac{x_1^2}{\mu_1} + \frac{y_1^2}{\mu_2} = \Delta \qquad (15)$$

$$\left\| \overrightarrow{BC} \right\|_2 = (x_{goal} - x_2)^2 + (y_{goal} - y_2)^2 \le x_{goal}^2 + y_{goal}^2 \qquad (16)$$

$$\left\| \overrightarrow{AC} \right\|_2 = (x_{goal} - x_1)^2 + (y_{goal} - y_1)^2 \le x_{goal}^2 + y_{goal}^2 \qquad (17)$$

Based on the the player A, the player B and the base camp b of player B are on the same plane. Given the player A's incomplete information strategy and their confrontation strategy with the player B, expression (18) represents the vectors formed by players A and B, respectively, with respect to the base camp b of player B. The modulus of the cross product of the two vectors can be expressed as the area of the parallelogram formed by the two vectors as sides. Meanwhile, the angle formed $\theta = \pi / 2$ by the vectors can be known from the game strategy under incomplete information.

$$f(x_1, x_2, y_1, y_2) = \left\| \overrightarrow{Bb} \times \overrightarrow{Ab} \right\|_2^2$$

$$= (\left| \overrightarrow{Bb} \right| \cdot \left| \overrightarrow{Ab} \right| \sin \frac{\pi}{2})^2 \qquad (18)$$

$$= [(x_{goal} - x_2)^2 + (y_{goal} - y_2)^2][(x_{goal} - x_1)^2 + (y_{goal} - y_1)^2]$$

Therefore, based on formulas(14), (15), (16),(17) and (18), we construct the Lagrangian function (19), and if $\delta, \varepsilon, \gamma, \eta$ none of them are zero, then

$$L(x_1, x_2, y_1, y_2, \delta, \varepsilon, \gamma, \eta) = f(x_1, x_2, y_1, y_2)$$
$$+ \delta(x_1^2 + x_2^2 + y_1^2 + y_2^2 - r^2)$$
$$+ \varepsilon(\mu_2 x_1^2 + \mu_1 y_1^2 - \mu_1 \mu_2 \Delta) \qquad (19)$$
$$+ \gamma((x_{goal} - x_2)^2 + (y_{goal} - y_2)^2 - (x_{goal}^2 + y_{goal}^2))$$
$$+ \eta((x_{goal} - x_1)^2 + (y_{goal} - y_1)^2 - (x_{goal}^2 + y_{goal}^2))$$

Equation (20) can be obtained from the joint constraint of Equation (19)

$$
\begin{cases}
\dfrac{\partial L}{\partial \delta} = x_1^2 + x_2^2 + y_1^2 + y_2^2 - r^2 = 0 \\[2mm]
\dfrac{\partial L}{\partial \varepsilon} = \mu_2 x_1^2 + \mu_1 y_1^2 - \mu_1 \mu_2 \Delta = 0 \\[2mm]
\dfrac{\partial L}{\partial \gamma} = (x_{goal} - x_2)^2 + (y_{goal} - y_2)^2 - (x_{goal}^2 + y_{goal}^2) = 0 \\[2mm]
\dfrac{\partial L}{\partial \eta} = (x_{goal} - x_1)^2 + (y_{goal} - y_1)^2 - (x_{goal}^2 + y_{goal}^2) = 0 \\[2mm]
\dfrac{\partial L}{\partial x_1} = \partial f / \partial x_1 + 2\delta x_1 + 2\mu_2 \varepsilon x_1 + 2\eta(x_1 - x_{goal}) \\[2mm]
\dfrac{\partial L}{\partial x_2} = \partial f / \partial x_2 + 2\delta x_2 + 2\gamma(x_2 - x_{goal}) \\[2mm]
\dfrac{\partial L}{\partial y_1} = \partial f / \partial y_1 + 2\delta y_1 + 2\mu_1 \varepsilon y_1 + 2\eta(y_1 - y_{goal}) \\[2mm]
\dfrac{\partial L}{\partial y_2} = \partial f / \partial y_2 + 2\delta y_2 + 2\gamma(y_2 - y_{goal}) \\[2mm]
\partial f / \partial x_1 = -2(x_{goal} - x_1)[(x_{goal} - x_2)^2 + (y_{goal} - y_2)^2] \\[2mm]
\partial f / \partial y_1 = -2(y_{goal} - y_1)[(x_{goal} - x_2)^2 + (y_{goal} - y_2)^2] \\[2mm]
\partial f / \partial x_2 = -2(x_{goal} - x_2)[(x_{goal} - x_1)^2 + (y_{goal} - y_1)^2] \\[2mm]
\partial f / \partial y_2 = -2(y_{goal} - y_2)[(x_{goal} - x_1)^2 + (y_{goal} - y_1)^2]
\end{cases} \qquad (20)
$$

So we can get the result as

$$
\begin{cases}
x_1 = \dfrac{x_{goal}(x_{goal}^2 + y_{goal}^2) + \eta x_{goal}}{(x_{goal}^2 + y_{goal}^2) + \delta + \mu_2 \varepsilon + \eta} \\[4mm]
y_1 = \dfrac{y_{goal}(x_{goal}^2 + y_{goal}^2) + \eta y_{goal}}{(x_{goal}^2 + y_{goal}^2) + \delta + \mu_1 \varepsilon + \eta}
\end{cases} \qquad (21)
$$

3.2 Global optimal solution validation of the agents

In the above argument,In this paper, we find the dynamic local optimal solution of player A in the 2D plane of the confrontation game, Then we want to verify the

global optimality of the local optimal solution of player A aggressive interference. When the objective function and constraint condition of the Lagrangian multiplier method are convex function, then the player A in this paper may give the global optimal solution to the convex optimization problem. Below, we need to verify that the objective function is convex and that the constraint conditions are convex sets.

3.2.1 The objective function is determined as a convex function

If $f(x_1, x_2, y_1, y_2) = [(x_{goal} - x_2)^2 + (y_{goal} - y_2)^2][(x_{goal} - x_1)^2 + (y_{goal} - y_1)^2]$ ,and $0 \le x_1 \le x_{goal}, 0 \le x_2 \le x_{goal}, 0 \le y_1 \le y_{goal}, 0 \le y_2 \le y_{goal}$ ,so $f(x_1, x_2, y_1, y_2)$ is it a convex function.

**Prove:** Set $A(x_2, y_2) = (x_{goal} - x_2)^2 + (y_{goal} - y_2)^2, B(x_1, y_1) = (x_{goal} - x_1)^2 + (y_{goal} - y_1)^2$ , Then the original function can be recorded as $f(x_1, x_2, y_1, y_2) = A(x_2, y_2) \cdot B(x_1, y_1)$ .Then The first partial derivative of each variable is shown in formula (21), (22), (23) and (24).

$$\frac{\partial f}{\partial x_1} = -2A(x_2, y_2)(x_{goal} - x_1) \tag{21}$$

$$\frac{\partial f}{\partial x_2} = -2B(x_1, y_1)(x_{goal} - x_2) \tag{22}$$

$$\frac{\partial f}{\partial y_1} = -2A(x_2, y_2)(y_{goal} - y_1) \tag{23}$$

$$\frac{\partial f}{\partial y_2} = -2B(x_1, y_1)(y_{goal} - y_2) \tag{24}$$

Then the second partial derivative of each variable is shown in equation (25), Equation (26), Equation (27) and Equation (28).

$$\frac{\partial^2 f}{\partial x_1^2} = 2A(x_2, y_2) \tag{25}$$

$$\frac{\partial^2 f}{\partial x_2^2} = 2B(x_1, y_1) \tag{26}$$

$$\frac{\partial^2 f}{\partial y_1^2} = 2A(x_2, y_2) \tag{27}$$

$$\frac{\partial^2 f}{\partial y_2^2} = 2B(x_1, y_1) \tag{28}$$

According to the above second derivative, it can be checked whether Hessian matrix is semi-positive definite matrix. The composition of Hessian matrix is shown in equation (29) and equation (30):

$$H(f) = \begin{vmatrix} \dfrac{\partial^2 f}{\partial x_1^2} & \dfrac{\partial^2 f}{\partial x_1 \partial x_2} & \dfrac{\partial^2 f}{\partial x_1 \partial y_1} & \dfrac{\partial^2 f}{\partial x_1 \partial y_2} \\ \dfrac{\partial^2 f}{\partial x_2 \partial x_1} & \dfrac{\partial^2 f}{\partial x_2^2} & \dfrac{\partial^2 f}{\partial x_2 \partial y_1} & \dfrac{\partial^2 f}{\partial x_2 \partial y_2} \\ \dfrac{\partial^2 f}{\partial y_1 \partial x_1} & \dfrac{\partial^2 f}{\partial y_1 \partial x_1} & \dfrac{\partial^2 f}{\partial y_1^2} & \dfrac{\partial^2 f}{\partial y_1 \partial y_2} \\ \dfrac{\partial^2 f}{\partial y_2 \partial x_1} & \dfrac{\partial^2 f}{\partial y_2 \partial x_2} & \dfrac{\partial^2 f}{\partial y_2 \partial y_1} & \dfrac{\partial^2 f}{\partial y_2^2} \end{vmatrix} \tag{29}$$

As a result of $\dfrac{\partial^2 f}{\partial x_i \partial x_j} = \dfrac{\partial^2 f}{\partial y_i \partial y_j} = 0(i \neq j)$ ,so

$$H(f) = \begin{bmatrix} 2A(x_2, y_2) & 0 & 0 & 0 \\ 0 & 2B(x_1, y_1) & 0 & 0 \\ 0 & 0 & 2A(x_2, y_2) & 0 \\ 0 & 0 & 0 & 2B(x_1, y_1) \end{bmatrix} \tag{30}$$

On account of $A(x_2, y_2) \geq 0, B(x_1, y_1) \geq 0$ ,Then all the eigenvalues in the Hessian matrix are non-negative, and the function $f(x_1, x_2, y_1, y_2)$ is convex.

3.2.2 Determination of the convex set of the constraints

Ⅰ.The constraint is $x_1^2 + x_2^2 + y_1^2 + y_2^2 = r^2$

According to the convex set definition: for any two points a, b points in this set point, and arbitrary $\theta \in [0,1]$, the linear combination $\theta a + (1-\theta)b$ should also be in this set.

Assume

$a = (a_1, a_2, a_3, a_4), b = (b_1, b_2, b_3, b_4)$ satisfied $a_1^2 + a_2^2 + a_3^2 + a_4^2 = b_1^2 + b_2^2 + b_3^2 + b_4^2 = r^2$

consider linear combinations

$$c_i = \theta a_i + (1-\theta)b_i, i = 1,2,3,4$$

As shown in the equation (31).

$$c_1^2 + c_2^2 + c_3^2 + c_4^2 = [\theta a_1 + (1-\theta)b_1]^2 + [\theta a_2 + (1-\theta)b_2]^2 + [\theta a_3 + (1-\theta)b_3]^2 + [\theta a_4 + (1-\theta)b_4]^2 \tag{31}$$

According to the Cauchy-Schwarz inequality as shown in Equation (32):

$$\forall x, y \in C, (\theta x + (1-\theta)y)^2 \leq \theta x^2 + (1-\theta)y^2 \tag{32}$$

Formula (33)

$$(\theta a_i + (1-\theta)b_i)^2 \le \theta a_i^2 + (1-\theta)b_i^2 \qquad (33)$$

So there is

$$c_1^2 + c_2^2 + c_3^2 + c_4^2 \le r^2\theta + (1-\theta)r^2 = r^2$$

So the constraints $x_1^2 + x_2^2 + y_1^2 + y_2^2 = r^2$ are not a convex set.

Ⅱ.The constraint is $\left\| \overrightarrow{BC} \right\|_2^2 = (x_{goal} - x_2)^2 + (y_{goal} - y_2)^2 \le x_{goal}^2 + y_{goal}^2$

This constraint represents the inner region of a circle. The interior and boundary of a circle form a typical convex set. This is because the segments that connect two points within the circle are completely located completely inside the circle.

Equal constraints $\left\| \overrightarrow{AC} \right\|_2^2 = (x_{goal} - x_1)^2 + (y_{goal} - y_1)^2 \le x_{goal}^2 + y_{goal}^2$ Also for convex sets.

To sum up,all the constraints are not all convex set,so this is not a convex optimization problem, so the best attack position of the player A is the local optimum.

4. Conclusion

This article primarily describes the confrontation between Player A and Player B in a strategic game. To address the problem of insufficient data that prevents the military chess deduction system from meeting real-time requirements, the external model-building method will serve as a necessary supplement to the big data support model. We aim to transform decision-making into a powerful auxiliary tool for commanders, with auxiliary model building creating latent advantages.

Based on the model consisting of three parts—the game triangle composed of Player A, the obstacle units arranged by Player B, and Player B's base camp point B.We establish the Lagrangian conditions and Lagrangian function for the game process between Player A and Player B, and find the locally optimal solution for Player A. Finally, the model constructed in this paper, along with the constraint conditions, provides an explanatory solution for the locally optimal solution.

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
