# OpenReview forum: "Position-Adaptive Optimization for Agent Confrontation Using the Game Triangle Plane Model"
_mathai.club/MathAI/2025/Conference — MathAI 2025 Oral_

### Official Review · Reviewer_ve66 · 2025-02-27
**Position-Adaptive Optimization for Agent Confrontation Using the Game Triangle Plane Model**

**Rating:** 3
**Confidence:** 5

**Review:**

The author does not follow the anonymity policy. The article does not match the template.

---

### Official Review · Reviewer_xMaf · 2025-02-27
**Hard to justify the relevance to the confernce topic**

**Rating:** 6
**Confidence:** 4

**Review:**

The mathematical content is interesting. However, it is hard to justify the relevance to the main topic of the conference (AI). The name of the author clearly indicated is not a big problem, as for me.

---

### Decision · Program_Chairs · 2025-03-08

**Decision:**

Accept (Oral)

**Comment:**

Your article has been accepted and you can make a presentation on the article. All articles will be sorted by rating and within the available conference places one author from each article will be invited. If there are not enough places, then you will either have the opportunity to present remotely or come at your own expense!